# Bringing Emotion Recognition Out of the Lab into Real Life: Recent Advances in Sensors and Machine Learning

**Stanisław Saganowski** 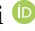

Department of Artificial Intelligence, Wrocław University of Science and Technology, 50-370 Wrocław, Poland; stanislaw.saganowski@pwr.edu.pl

**Abstract:** Bringing emotion recognition (ER) out of the controlled laboratory setup into everyday life can enable applications targeted at a broader population, e.g., helping people with psychological disorders, assisting kids with autism, monitoring the elderly, and general improvement of well-being. This work reviews progress in sensors and machine learning methods and techniques that have made it possible to move ER from the lab to the field in recent years. In particular, the commercially available sensors collecting physiological data, signal processing techniques, and deep learning architectures used to predict emotions are discussed. A survey on existing systems for recognizing emotions in real-life scenarios—their possibilities, limitations, and identified problems—is also provided. The review is concluded with a debate on what challenges need to be overcome in the domain in the near future.

**Keywords:** emotion recognition; field studies; wearables; signal processing; machine learning; review

## 1. Introduction

Emotions are a basic component of our life, just like breathing or eating. They are also responsible for a majority of our decisions [1]. For that reason, emotion recognition is an important and profitable research problem. The ability to recognize emotions can help in emotion-based disorders [2], autism [3,4], monitoring our well-being [5,6] and mental health [7], controlling stress [8], human–computer interaction [9], recommendation systems [10,11], and computer games [12,13]. At the same time, emotion recognition is a very ambitious task, as it connects several disciplines, i.e., psychology, electronics/sensors, signal processing, and machine learning; see Figure 1a.

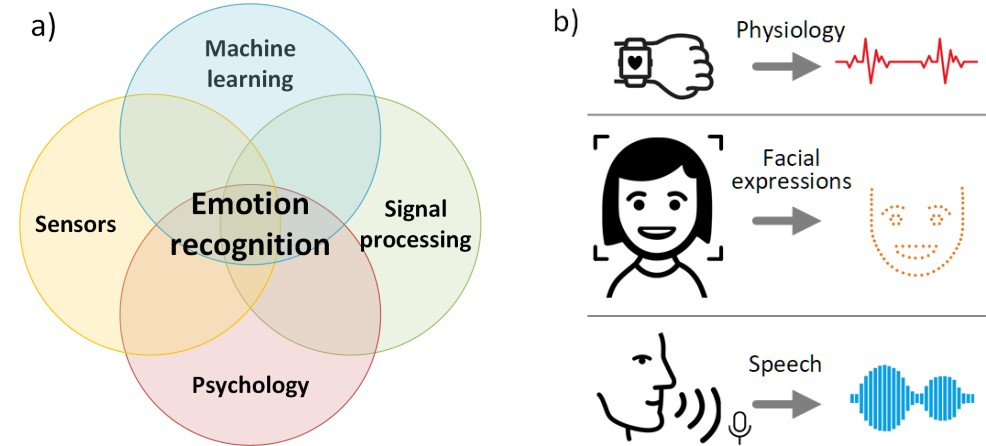

**Figure 1.** (**a**) The interdisciplinary background of emotion recognition; (**b**) modalities used to recognize emotions.

For years, psychologists have analyzed people and the processes taking place between the affective situation and human reaction and behavior. This allowed them identify

different perspectives on emotions [14] and define various models such as basic emotion models [15–17], appraisal models [18,19], psychological construction models [20,21], and social construction models [22,23]. As a consequence, there is no single commonly accepted definition of emotions.

Electronics created small and effective sensors that capture images/videos, record audio, and collect physiological signals, as seen in Figure 1b. Signal processing allows us to efficiently process the collected data, improving their quality and obtaining informative features. Machine learning, in turn, enables us to analyze vast amounts of data, automatically extract patterns, create affective states' representations, and thus recognize or even predict emotions or other affective states.

This work provides the state of the art in the fields of wearable sensors, signal processing, and machine learning models and techniques, adequate for the emotion recognition from speech, facial images, and physiological signals. The crucial aspects of each domain are explained with examples. Rather than providing the details of each method/system, their general idea is presented, positioned in the context of the problem, and the reference to the source article is provided to keep the article concise.

The primary focus of this review is on emotion recognition from the physiological signals because it can be performed continuously in everyday life using wearables, as opposed to facial and speech emotion recognition. However, the latter modalities are also covered in this work. The abbreviations used in the article are listed in the Acronyms section at the end of the paper.

## 2. Sensors and Devices

Different modalities can be utilized to perform emotion recognition, but first, the appropriate sensors have to be utilized to capture these modalities.

### 2.1. Emotion Recognition from Physiology

Among the physiological signals, the EEG provides the highest accuracy of emotion recognition [24–26]. However, due to the EEG electrodes sensitivity, at this point, it is impossible to utilize EEG in the field. Therefore, researchers focused on wearable devices, which provide various biosignals and other environmental data [27]. Primarily, ECG, BVP, and EDA signals are used [5,9,12]. Those signals can be further enriched with ACC, GYRO, SKT, and RSP [2,7,8,28,29], as well as with UV, GPS, and MIC data [30].

Numerous devices for obtaining physiological signals are available on the market. Precise medical-grade devices, such as the BioPac MP160 and the ProComp Infiniti, can be used for laboratory studies. They are large, wired, and complicated, but they are also handled by a trained technician. Devices for studies in the field, however, must be small, wearable, and easy to operate by the participant. There are many types of wearables to choose from, e.g., smart rings, wristbands, smart bands, fit bands, smartwatches, armbands, chest straps, chest patches, headbands, smart glasses, smart clothes, and more.

The choice of a particular type and model depends on the study we plan to perform [31]. For instance, devices monitoring cardiac and electrodermal activity are the best choice for emotion recognition in the field. Comfortable EEG headbands can be used for meditation studies. Dedicated devices are necessary for analyzing sleep, while commercial smartwatches are sufficient for monitoring physical activity. When selecting a device for studies in the field, the following factors should be taken into consideration:

- What physiological signals do we need to collect? Does the device provide them in a raw format?
- What is the signal recording frequency? Is it appropriate for our problem?
- How portable is the device? What types of physical activities does the device have to handle?
- How are the data obtained from the device? By cable or directly transferred to the cloud? Do we need to integrate the existing study system with the device?
- How convenient and easy to use is the device? How long should the battery last?

Table 1 provides an overview of various types of wearables, their sensors, and physiological signals and other data they provide. The author manually investigated the devices marked with *. For other devices, the official producers' or other external information were used. Most devices included in the table were introduced over the past five years. Additionally, a few older devices, which were, or still are, frequently utilized in affective studies, are covered, e.g., Empatica E4 released in 2015 and BodyMedia SenseWear introduced in 2003.

Smartwatches seem to be the most convenient device for the participants and offer some extra features. Moreover, people already have a long history of wearing watches and, thus, are more willing to accept such devices. As for the downside of smartwatches, their HR and HRV readings are not very precise when participants are in motion [32]. More accurate measurements can be obtained with chest straps [33], which, in turn, are not very comfortable. Perhaps, a compromise would be a chest patch such as Vital Patch, although it can only run on a battery for a limited time and still requires more validation studies.

Other devices analyzed within this work were Emotiv Epoc+*, NeuroSky, emWave2, Honor Band 4*, Xiaomi Mi Band 5*, Xiaomi Mi Band 3*, Polar A370*, Fitbit Blaze*, Samsung Galaxy Fit E, Samsung Gear Live, Philips DTI-2, Sony SmartBand 2, and 20 others. They do not offer access to the data or have other disadvantages; hence, they were not included in Table 1.

### 2.2. Facial Emotion Recognition

Emotion recognition from facial expressions requires a camera to obtain the face image. Ideally, the face would be turned toward the camera, without any obstacles (hair, eyeglasses, and face mask), and the image would be captured in high resolution and good lighting conditions. Such criteria are easy to fulfill in a laboratory environment but very troublesome in daily life. It is possible to use a smartphone or laptop camera, but this limits the emotion recognition to situations when the subject is in front of the camera. McDuff et al. performed a field study utilizing laptop cameras to analyze facially expressed affection [34]. They recorded about 120 people for approximately eight days between 7 a.m. and 7 p.m. and found out that people were in front of the computer on average for 19% of this time. However, there are still use-cases in which ER from the face can be helpful, e.g., in online learning to ensure students are focused and engaged instead of being bored or frustrated [35].

### 2.3. Speech Emotion Recognition

In the case of ER from speech, the microphone is responsible for recording audio signals. Again, it is easier to obtain clean, undisturbed audio without background noise in a laboratory setup. It is also possible at home when we interact with speaker-based home assistants, such as Alexa or Google Home. Outdoor, however, the environmental sounds and other people's utterances are most likely captured alongside the main speaker's voice. Nevertheless, speech emotion recognition in the field might still be possible with appropriate filtering. For example, Lu et al. proposed the StressSense, a smartphone application that can detect stress in diverse real-life conversational situations, both indoor and outdoor [36].

**Table 1.** The recent commercially available wearable devices, their sensors measuring physiological signals, and the data they provide; (*) marks wearables tested by the author.

| Device | Type | Release Date | Sensors | Physiological Raw Signals | Other Data |
|---|---|---|---|---|---|
| Apple Watch 7 | Smartwatch | October 2021 | ACC, AL, ALT, BAR, ECG, GPS, GYRO, MIC, PPG | BVP, ECG, SpO2 | ACC, AL, ALT, BAR, BP, GPS, GYRO, HR, MIC, RSP, SKT, STP |
| Fitbit charge 5 | Smartband | September 2021 | ACC, BAR, GPS, GYRO, PPG | - | ACC, BAR, GYRO, HR |
| Samsung Galaxy Watch 4* | Smartwatch | August 2021 | ACC, AL, BAR, ECG, GPS, GYRO, MIC, PPG | BVP | ACC, AL, BAR, GPS, GYRO, HR, MIC, PPI, STP |
| Huawei Watch 3 | Smartwatch | June 2021 | ACC, AL, BAR, GPS, GYRO, MAG, MIC, PPG, TERM | - | ACC, AL, BAR, GPS, GYRO, HR, MAG, MIC, SKT, STP |
| Fitbit Sense | Smartwatch | September 2020 | ACC, AL, ALT, GPS, GYRO, MIC, PPG, TERM | - | ACC, BAR, GYRO, HR |
| Samsung Galaxy Watch 3* | Smartwatch | August 2020 | ACC, AL, BAR, ECG, GPS, GYRO, MIC, PPG | BVP | ACC, AL, BAR, GPS, GYRO, HR, MIC, PPI, STP |
| Apple Watch 5* | Smartwatch | September 2019 | ACC, BAR, ECG, GPS, GYRO, MIC, PPG | - | ACC, BAR, CAL, GPS, GYRO, HR, MIC, STP |
| Fossil Gen 5* | Smartwatch | August 2019 | ACC, AL, ALT, GPS, GYRO, MIC, PPG | BVP | ACC, AL, ALT, GPS, GYRO, HR, MIC, STP |
| Garmin Fenix 6X Pro | Smartwatch | August 2019 | ACC, AL, ALT, GPS, GYRO, PPG, SpO2 | BVP, SpO2 | ACC, AL, ALT, GPS, GYRO, HR, STP |
| Samsung Galaxy Watch* | Smartwatch | August 2019 | ACC, AL, BAR, GPS, GYRO, MIC, PPG | BVP | ACC, AL, BAR, GPS, GYRO, HR, MIC, STP |
| Polar OH1 | Armband | March 2019 | ACC, PPG | BVP | ACC, PPI |
| Garmin HRM-DUAL | Chest strap | January 2019 | ECG | ECG | RRI |
| Muse 2* | Headband | January 2019 | ACC, EEG, GYRO, PPG, SpO2 | BVP, EEG, SpO2 | ACC, GYRO, HR |
| Fitbit Charge 3* | Fitband | October 2018 | ACC, ALT, GYRO, PPG | - | ACC, ALT, HR |
| Garmin VivoActive 3 Music* | Smartwatch | June 2018 | ACC, BAR, GPS, GYRO, PPG | - | ACC, CAL, HR, PPI, RSP, STP |
| Oura ring* | Smart ring | April 2018 | ACC, GYRO, PPG, TERM | - | HR, PPI, SKT |
| Moodmetric* | Smart ring | December 2017 | ACC, EDA | EDA | STP |
| DREEM | Headband | June 2017 | ACC, EEG, PPG, SpO2 | BVP, EEG, SpO2 | ACC, HR |
| Aidlab | Chest strap | March 2017 | ACC, ECG, RSP, TERM | ECG, RSP, SKT | Activities, HR, HRV, STP |
| Polar H10 | Chest strap | March 2017 | ACC, ECG | ECG | ACC, RRI |
| VitalPatch | Chest patch | March 2016 | ACC, ECG, TERM | ECG, SKT | HR, RRI, RSP, STP |
| Emotiv Insight | Headband | October 2015 | ACC, EEG, GYRO, MAG | EEG | ACC, GYRO, MAG |
| Empatica E4* | Wristband | 2015 | ACC, EDA, PPG, TERM | BVP, EDA, SKT | ACC, HR, PPI |
| Microsoft Band 2 | Smartband | October 2014 | ACC, AL, ALT, BAR, EDA, GYRO, PPG, TERM, UV | BVP, EDA, SKT | ACC, AL, ALT, BAR, CAL, GYRO, HR, PPI, STP, UV |
| BodyMedia SenseWear | Armband | 2003 | ACC, EDA, TERM | EDA, SKT | ACC |

## 3. Signal Processing and Transformation

The goal of signal processing is to eliminate interference and distortions from physiological signals which are not related to emotional characteristics and might bias the predictive ability of the models. The possible sources of signal artifacts are peripheral electromagnetic fields impacting electronic circuits within wearables, mixing of signals generated by different organs (e.g., the heart, brain, or muscle electrical activities), temporary deactivation or saturation of sensors, electronic noise generated inside the circuits, body motions, and movements and adjustments in sensor contact with the skin. The common approaches to reducing signal distortions are based on the known properties of the physiological signals. For example, the amplitude of the ECG signal should not exceed values 0.5–5 mV, whereas the ECG frequency spectrum is usually within 0.05–150 Hz values. On the other hand, the BVP amplitude is usually within 20–300 mmHg and the BVP frequency spectrum within 0–15 Hz. Alternatively, the data-driven approach can be applied. We can distinguish four main groups of methods for removing artifacts: (1) filtration and smoothing; (2) decomposition and removing undesirable components; (3) normalization; and (4) winsorization.

According to the Nyquist–Shannon sampling theorem, before converting the processed voltage into a digital signal, initial antialiasing filtering should be performed. The application of additional filtering (both analog and digital) and smoothing emphasize desired frequency components while reducing unwanted ones. High-pass filters, for example, eliminate slowly wandering components; low-pass filters minimize high-frequency disturbances; pass-band filters focus on specific frequency ranges; and signal-smoothing techniques prioritize low-frequency components.

Decomposition facilitates the separation of desired and undesired components. The wavelet transform (WT), which is available in continuous (CWT), discrete (DWT), and fast (FWT) variations, turns a nonstationary input onto the coefficient components of various scales (also connected to frequency ranges) by using a hand-picked family of wavelets to detect local correlations. The first component, in most cases, is a measurement noise or an electromyogram anomaly that can be eliminated afterward. Independent component analysis (ICA) is yet another method for distinguishing independent input signals (mixed while recorded by multiple sensors).

Normalization allows us to retain the energy of signals at a consistent level, especially when data from multiple devices are merged. Interpolation replaces damaged parts of the signal while accounting for the statistical features of some or all neighboring data. Lastly, winsorization reduces extreme values to reduce the effect of possibly spurious outliers.

Lee et al. showed that the application of the signal processing and transformation methods, i.e., ICA, fast Fourier transform (FFT), truncated singular value decomposition, can reduce motion artifacts from the BVP signal recorded with a wearable device, resulting in more precise HR readings, even during intense exercises [37,38]. Masinelli et al. proposed the SPARE (spectral peak recovery) algorithm for BVP signals pulse-wave reconstruction [39]. The SPARE performs signal decomposition with singular spectrum analysis, spectral estimation with sparse signal reconstruction and FFT, harmonic relationship estimation, and reconstruction by applying a narrow bandpass filter to the resulting components and summing them up. The validation performed by the authors showed a 65% improvement in the detection of different biomarkers from the BVP signal.

Signal processing and transformation can also be helpful in feature extraction. Feng et al. utilized wavelet-based features to classify emotions from the EDA signal [3]. They compared four *mother* wavelet candidates, i.e., Daubechies, Coiflets, Symlets, and C-Morlet and found the C-Morlet to be the best. Zhao et al. applied FFT to obtain subbands of BVP and HRV signals collected with Empatica E4 device [40]. They also used dynamic threshold difference (DTD) to obtain HRV from BVP.

## 4. Machine Learning Models and Techniques

Emotion recognition usually involves a massive amount of data, e.g., speech or physiological signals sampled with high-frequency, facial images (or videos) captured in a high resolution. Processing such extensive sequential data requires complex structures, adequate methods, and significant computing power. Currently, the best approach is to utilize deep neural network architectures capable of reflecting temporal dynamic behavior of affective data, thus providing rich representational possibilities. The network architectures explicitly designed to capture the temporal information of data are the recurrent neural networks (RNNs). Besides emotion recognition, RNNs are used in problems such as time-series prediction, speech recognition, machine translation, and robot control. However, in their classic design, the RNNs have gradient vanishing and exploding problems. Architectures introducing gates (gated recurrent unit—GRU, long short-term memory—LSTM) and shortcuts (residual networks—ResNet) were proposed to overcome these problems.

Another deep neural network architecture helpful in emotion recognition is convolutional neural networks (CNNs). CNNs are excellent in image vision problems; hence, they are convenient in face detection and analysis tasks and can also be applied to signals' spectrograms. A common approach is to mix and combine several architectures to benefit from their advantages.

In order to develop machine learning models capable of emotion recognition, we need data obtained from sensors to be labeled with affective states. This is usually accomplished by asking the participants to provide a self-assessment. The labeled dataset is commonly divided into three parts: one for training the model, another one for validating/optimizing the model, and the last one for testing the performance of the model. The deep learning models may require hundreds, thousands, or tens of thousands of samples to perform well. By contrast, the affective datasets are most often sparse and contain up to 50 participants (up to a few thousand samples).

The following sections describe the most popular deep learning architectures for emotion recognition problems and several techniques for better model creation and adjustment. Additionally, some of the recent emotion recognition studies utilizing state-of-the-art deep learning architectures are summarized in Table 2 and briefly described in the text. In most cases, the performance of emotion recognition is also reported. Please note that it is provided only to give some general idea of what level of performance is achieved in the literature. The results should not be compared without carefully analyzing the details of each work. Studies on emotion recognition can be different in many ways, e.g., they may vary in: (1) the dataset used; (2) the emotional model applied; (3) the machine learning approach adopted; (4) the number of classification classes and their distribution; (5) the validation strategy (e.g., user-independent vs. user-dependent); (6) whether the results are provided for train, validation, or test set; and (7) the performance quality measure.

**Table 2.** The recent emotion recognition studies utilizing state-of-the-art deep learning architectures.

| Modality | Reference | Architecture | Classification/Regression Problem Considered |
|---|---|---|---|
| Phys. signals | Awais et al., 2021 [41] | LSTM | 4-class: amused, bored, feared, relaxed |
| | Dar et al., 2020 [42] | CNN + LSTM | 4-class: high/low arousal/valence |
| | Song et al., 2020 [43] | CNN | 2 × 3-class: calm/medium/excited arousal; unpleasant/neutral/pleasant valence |
| | Tizzano et al., 2020 [44] | LSTM | 3-class: happy, neutral, sad |
| | Kanjo et al., 2018 [30] | CNN + LSTM | 5-class: level of valence (from 0 to 4) |
| | Nakisa et al., 2018 [25] | LSTM | 4-class: high/low arousal/valence |
| | Santamaria-Granados et al., 2018 [45] | CNN | 2 × binary: high/low arousal/valence |
| Facial expression | Li and Lima, 2021 [46] | ResNet-50 | 7-class: angry, disgusted, fearful, neutral, happy, sad, surprised |
| | Sepas-Moghaddam et al., 2020 [47] | VGG16 + Bi-LSTM + attention | 4-class: angry, happy, neutral, surprised |
| | Efremova et al., 2019 [48] | ResNet-20 | 5-class: positive, weak positive, neutral, weak negative, negative |
| | Cheng et al., 2017 [49] | FCN + CNN | regression: valence value |
| | Bargal et al., 2016 [50] | ResNet-91 with 2 × VGG | 7-class: angry, disgusted, fearful, neutral, happy, sad, surprised |
| Speech | Fan et al., 2021 [51] | PyResNet: ResNet with pyramid convolution | 4-class: angry, neutral, happy, sad |
| | Wang et al., 2020 [52] | dual-sequence LSTM | 4-class: angry, neutral, happy, sad |
| | Yu and Kim, 2020 [53] | attention-LSTM-attention | 4-class: angry, neutral, happy, sad |
| | Zhang et al., 2019 [54] | FCN-attention | 4-class: angry, neutral, happy, sad |
| | Zhao et al., 2019 [55] | attention-Bi-LSTM + attention-FCN | 4-class: angry, neutral, happy, sad; 5-class: angry, emphatic, neutral, positive, resting |
| | Li et al., 2019 [56] | ResNet | 7-class: angry, bored, disgusted, fearful, neutral, happy, sad |
| Visual + phys. signals | Gjoreski et al., 2020 [57] | StresNet, CNN, LSTM | binary (driver distraction) |

### 4.1. Residual Networks

In 2015, He et al. proposed the ResNet [58] architecture, which facilitates creating very deep neural networks without concerns about performance and the vanishing gradient issue. All thanks to shortcuts that omit one or more layers. Later, He et al. refined ResNet with a pre-activation residual block, enabling even deeper networks [59]. The ResNet takes an image as the input and includes convolution layers. It is especially helpful in image processing problems, such as face recognition and landmark detection. It can also be applied to physiological or speech data once the signal is represented as an image or its features, e.g., power spectral density or other characteristics from the time or frequency domains.

For instance, Li and Lima used ResNet-50 (containing 50 layers) to perform facial expression recognition [46]. They achieved average accuracy of 95 ± 1.4% in recognizing seven classes: happy, sad, fearful, angry, surprised, disgusted, and neutral. Efremova et

al. managed to embed a 20-layer ResNet model on a mobile device to perform real-time face and emotion recognition [48]. They distinguish positive, weak positive, neutral, weak negative, and negative classes and obtained 87% of accuracy on the test set. Bargal et al. concatenated 91-layer ResNet architecture with two VGG networks [60] to perform emotion recognition from videos [50]. They argue that this provides some form of regularization. They identify six basic emotions (anger, disgust, fear, happiness, sad, and surprise) and neutral state with 57% of accuracy on the test set.

Li et al. utilized ResNet to investigate speech emotion recognition [56]. They replaced the last layer with classifiers and observed the best results with the SVM classifier (on average 90% accuracy). They used EMODB [61], a publicly available dataset that offers about 500 utterances with anger, boredom, disgust, fear, happiness, sadness, and neutral attitude. Fan et al. proposed PyResNet to classify emotions from speech [51]. The name comes from modifying the second layer of the ResNet with pyramid convolution, which should reduce the issue of uncertain time position of accurate speech information. They collected almost 150,000 utterances labeled with numerous emotions (anger, happiness, sadness, disappointment, boredom, disgust, excitement, fear, surprise, normal, and others). PyResNet based on the ResNet152 achieved 62% of weighted accuracy on the test set in speech classification. Their dataset is publicly available.

Various modifications of the ResNet architecture were proposed. Gjoreski et al. introduced StresNet [57] (originally named STRNet), which aims to capture a temporal and spectral representation of the signal. In their work, StresNet and other deep architectures (CNN and LSTM) were used to assess the possibility of monitoring driver distractions from physiological and visual signals. They obtained 75% of F-measure with StresNet on the test set and the participant independent scenario in recognizing whether distraction occurred (binary classification).

*4.2. Long Short-Term Memory*

To mitigate the gradient vanishing and exploding problems of the RNNs, the long short-term memory [62] (LSTM) architecture has been proposed. The LSTM cells contain *forget gates* that control how much information is passed on . This also allows capturing long-term temporal dependencies of the data, making LSTMs very attractive in sequential processing tasks, e.g., reasoning affective state from the physiological signals. The LSTM input layer requires three-dimensional data that refer to sequence (samples), time steps, and features.

Awais et al. applied LSTM architecture to classify four emotions (amusement, boredom, relaxation, and fear) based on physiological signals [41]. Their multimodal approach achieved results above 93% of F-measure value. They used CASE [63], a publicly available dataset with a rich spectrum of signals, i.e., ECG, BVP, GSR, RSP, SKT, and EMG. Tizzano et al. used LSTM to recognize happy, sad, and neutral affective states in subjects that were listening to music or watching a short movie [44]. They claimed to obtain 99% accuracy. The physiological data (heart rate, 3-axis acceleration, and angular velocity) were obtained with wearable devices—Samsung Gear 2 and Polar H7. Nakisa et al. focused on the LSTM hyperparameter optimization in high/low arousal/valence classification task [25]. As an input, they used EEG and BVP physiological signals recorded with wearable devices—the Emotiv Insight headset and Empatica E4 wristband. They found differential evolution to be the best optimization algorithm. On average, they accomplished $67 \pm 9.3\%$ of accuracy in 4-class classification problem. Dar et al. utilized CNN and LSTM to classify high/low arousal/valence from EEG, ECG, and EDA signals [42]. They claimed 91% and 99% accuracy on DREAMER and AMIGOS datasets, respectively. Wang et al. introduced a dual-sequence LSTM architecture that is capable of processing two sequences of data simultaneously [52]. In their case, the mel-spectrogram focused on the time axis, and the mel-spectrogram favoring frequency axis is considered to classify utterances into happy, angry, sad, and neutral states. They achieved $72.7 \pm 0.7\%$ in mean weighted accuracy.

The LSTM architecture can also be used in the bidirectional version—Bi-LSTM, which has two recurrent layers. This allows one to propagate information in two directions (forward and backward) and is helpful for sequencing sequence tasks. Moreover, the LSTM utilized as encoder–decoder is often extended with the *attention mechanizm*. It allows passing the information from all the encoder nodes to all the decoder nodes, in contrast to the classic scheme where only the last encoder node forwards the condensed information to the first decoder node.

Sepas-Moghaddam et al. used VGG16 and bidirectional version of the LSTM, further enhanced with the attention mechanism to recognize happy, angry, surprised, and neutral states from facial images [47]. They obtained 87.6±5.4% accuracy in subject-specific and 80.3 ± 9% in subject-independent validation. Yu and Kim proposed an attention-LSTM-attention model for classifying happy, angry, sad, and neutral states from 5-second speech samples [53]. They first generated new features with the attention mechanism and then applied bidirectional LSTM, and finally perform weighted pooling by an attention mechanism. Their approach achieved 67.7 ± 3.4% in weighted accuracy. Zhao et al. developed a model that combines attention-based bidirectional LSTM with attention-based fully convolutional networks to obtain spatial-temporal features from the speech spectrogram [55]. They validated their architecture on two datasets to classify (1) angry, happy, sad, and neutral states, and (2) anger, emphatic, neutral, positive, and rest states. They achieved 67% and 49% of unweighted accuracy, respectively, and concluded that deep representations are competitive to feature representations in speech emotion recognition tasks.

### 4.3. Convolutional Neural Networks and Fully Convolutional Networks

The idea of using convolutional neural network (CNN) to handwritten character recognition was proposed in the 1990s [64], but only with graphics processing units (GPUs) implementation could CNN be applied more broadly and effectively. The CNN performs excellent at image classification problems due to its pattern recognition abilities. The CNN comprises (1) convolutional layers that apply filters to extract features, (2) pooling layers that reduce the dimensionality, and (3) fully connected layers that aggregate information from final feature maps.

The Fully Convolutional Network [65] (FCN) architecture was proposed for semantic segmentation in images. In contrast to CNN, FCN allows input images to have various resolutions. Additionally, the FCN does not flatten the convoluted image and instead keeps it as $1 \times 1 \times depth$ layer to proceed with deconvolution (upsampling) to the original size. The idea is similar to the encoder (convolution part) and decoder (deconvolution), where we lose some information in favor of general representation (e.g., semantic segmentation without details about segments).

In emotion recognition problems, the CNN and FCN architectures are mostly used for facial analysis in photos and videos. Once we convert the signals into spectrograms or input as a vector, they can also be used for emotion recognition from speech or physiological signals.

Cheng et al. proposed an architecture based on FCN and CNN to assess valence value from facial video recordings [49]. They designed the super-resolution FCN architecture to ensure the robustness of the model in case the video is of low quality or low bit rate, e.g., a video being streamed in real-time. In the best setup, they reported 0.121 of RMSE. Zhang et al. created attention-based FCN to classify speech into four classes, i.e., happy, sad, angry, and neutral [54]. Applying FCN allowed them to use speech fragments of various lengths without segmentation, which would have been necessary in the case of CNN. No need for segmentation also means there is no risk of losing potential information. The authors claimed that using the attention mechanism makes it easier for the model to identify the region of the speech spectrogram indicating the affective state. They accomplished 70% in weighted accuracy. Santamaria-Granados et al. utilized CNN to perform 2-level arousal and 2-level valence classification using ECG and EDA physiological signals [45]. They achieved 76% accuracy for arousal and 75% for valence. Song et al. utilized CNN to perform

3-level arousal and 3-level valence classification using EEG, ECG, EDA, Resp, and SKT physiological signals [43]. They attained 62% and 58% accuracy for arousal and valence, respectively. Kanjo et al. enhanced the CNN architecture with the temporal dimension of the data by appending the LSTM cell after the fully connected layer [30]. The classification task was a 5-level valence. As an input, they used the tensor of HR, EDA, Temp, ACC, and environmental data, collected with wristband Microsoft Band 2, in a real-life scenario. They obtained 95% of accuracy (95% F-measure, 0.291 RMSE) on the test set.

### 4.4. End-to-End Deep Learning Approach

In the classical feature-based approach to model creation, we manually extract features from the available data. This, however, requires expert domain knowledge to (1) preprocess signal or image (e.g., signal: wavelet/Fourier transform, spectral analysis; image: adjusting resolution, removing Gaussian noise), (2) obtain relevant features (e.g., signal: morphology, statistical, and nonlinear measures; image: facial landmarks, action units), and (3) select only valuable ones. Even then, it is possible that we do not capture all the complex dependencies and characteristics of the affective modalities. Thanks to more advanced deep neural network architectures and more powerful computers, it has become possible to create *end-to-end* networks that retrieve complex features from the raw signal or image themselves.

Dzieżyc et al. compared ten deep neural network architectures (ResNet, StresNet, FCN, MLP, Encoder, Time-CNN, CNN-LSTM, MLP-LSTM, MCDCNN, and Inception) in the end-to-end affect recognition from physiological signals task [66]. They found that the networks perform better when arousal and valence classes are well separated (strongly differentiating stimuli) and worse when the class boundary is blurred. The best results were achieved with FCN, ResNet, and StresNet. Schmidt et al. utilized CNN to build an end-to-end architecture [67]. They provided BVP, EDA, ACC, and SKT physiological signals as input. Their end-to-end model performed better ($45.5 \pm 1.8\%$ F-measure) than the feature-based model ($43.8 \pm 2.0\%$ F-measure) for high/low valence/arousal classification. Zhao et al. proposed an end-to-end visual–audio attention network (VAANet), which consists of 2D and 3D ResNets, and several CNNs with attention [68]. Their architecture can recognize multiple emotions from videos. The validation on the VideoEmotion-8 (eight classes: anger, disgust, fear, sadness, anticipation, joy, surprise, and trust) and Ekman-6 (six classes: anger, disgust, fear, sadness, joy, and surprise) benchmarks indicated state-of-the-art performance—54.5% and 55.3% accuracy, respectively. Sun et al. used residual CNN as end-to-end architecture in the speech emotion recognition task [69]. A part of the architecture is responsible for recognizing gender. Sun validated his model on three datasets with different languages (Mandarin, English, and German) and outperformed models based on the classical features, spectrograms, and other non-end-to-end deep learning solutions, achieving up to 90.3% accuracy. Harper and Southern fed the interbeat interval (IBI) signal into CNN and Bi-LSTM separately and concatenated networks' output to classify valence into low or high [70]. They performed very little processing, i.e., extracted IBI from HR and applied z-scoring; hence, their approach can be considered end-to-end. Their architecture outperformed the feature-based model obtaining up to 90% accuracy and 88% in F-measure value.

### 4.5. Representation Learning

Most of the works presented in this article use the advantages of learning data representation. Usually, it is capturing/extracting features from the data. However, when learning deep networks, the representation of signals or images can have more benefits, such as improving signal quality, generating missing samples or the whole signal, and translating one signal into another.

A denoising autoencoder (DAE), facilitating encoder–decoder architecture with convolutional layers, can be used to obtain a clean uncorrupted signal from the corrupted input

signal. Chiang et al. compared the noise reduction abilities of DAE, CNN, and FCN for ECG signal and found DAE to be the best [71].

The generative adversarial networks (GANs) are great for generative modeling. They consist of two neural networks that compete against each other. The generator network produces artificial samples from a random distribution, whereas the discriminator network tries to recognize which samples are real and which are artificial. Adib et al. investigated five different models from the GAN family for ECG signal generation task [72]. They compared their effectiveness in *normal* cardiac cycles and found that the classic GAN performed best. Sun et al. utilized LSTM and GAN networks to generate abnormal ECG signal [73].

Samyoun et al. applied representation learning to translate signals/features from the wrist device into the signals/features from the chest device [74]. They used GANs to translate homogeneous signals (EDA from the wrist into EDA from the chest), Bi-LSTM to translate heterogeneous signals (BVP from the wrist into ECG and Resp from the chest), and MLP to translate features of uncorrelated signals (all wrist features into EMG from chest features). They showed that stress detection with translated features has a competitive performance to stress detection with the original signals. This opens the possibility of using more user-friendly wrist devices instead of inconvenient chest devices in real-life affective studies.

*4.6. Model Personalization*

An important aspect to consider in the emotion recognition task is model generalization and personalization. On the one hand, we would like to have a predictive model that can be applied irrespectively to the participants' demographic and physiological characteristics. This would allow recognizing emotions in people that did not provide any prior data or are interacting with the system for the first time. On the other hand, people are very different in terms of psychological and physiological elements when it comes to expressing and perceiving emotions. Hence, a personal model might perform better than the general one.

The most straightforward approach to personalization is creating a separate model for each user. However, it requires a large number of personal samples. Udovičić et al. achieved better results with the per-person model than with the general model in the arousal and valence classification from EDA and BVP signals [75]. Tian et al. clustered participants into five groups based on their personality traits [76]. Such per-group personalization was enough to improve the model's accuracy in relation to the general model for arousal and valence classification using the EEG signal by 10.2% and 9.1%, respectively. Taylor et al. proposed a multitask learning neural network architecture that enables model personalization in the assessment of the mood, stress, and health of individuals in the next day[77]. Rather than creating a separate model for each user, the multitask ability enables personalization. They obtained up to 82.2% accuracy (0.818 AUC). Can et al. demonstrate that creating a separate model for each participant outperformed per-group personalization and general model in stress classification task [78]. They achieved a gain of up to 17 percentage points.

**5. Existing Systems**

The iMotions is a platform with the broadest spectrum of services targeted at affective research and analysis for scientific and industry clients [79]. For both types of clients, the platform is paid. According to the pricing plan, it is very expensive – for academics, it is ca. $3.5k per module, and a module is usually related to a single modality. The software can collect and synchronize data from many sensors, among others: EDA, EEG, ECG, EMG, and eye-tracking and facial recognition devices. The iMotions offer integration with the most popular devices on the market produced by Emotiv, Empatica, Shimmer, and Tobii. It also includes a signal processing module that can perform basic processing such as obtaining HR and HRV from ECG, detecting peaks in EDA, obtaining PSD from EEG, and others.

Moreover, a mobile module is available that facilitates conducting field studies. However, there are yet no use-cases to validate this module. At the current stage, the most valuable benefits of the platform are collecting self-assessments and data from various devices and performing analyses. With more features (especially signal processing algorithms), the platform would gain attractiveness.

Gloor et al. developed the *Happimeter*, an Android application for monitoring emotions in daily life [80]. The system requires a smartwatch that measures HR and collects self-assessments and location data. Once an appropriate amount of data is collected, the system starts predicting the user's mood. The Happimeter offers interesting insights, such as who or what location makes us (un)happy, active, or stressed. It also includes the social aspect—we can build a network of friends and monitor their mood. The application was developed for Wear OS smartwatches and is publicly available. The advantage of the Happimeter is that it creates a personalized model to perform predictions. On the other hand, the HR is collected infrequently (the exact frequency is not specified) and as a simple averaged value instead of HRV, BVP, or other continuous, rich signal. As a result, the Happimeter is more likely to recognize the overall long-term mood rather than the specific emotions experienced at a specific moment. Moreover, it can only measure three levels of mood, activity, and stress. The researchers give a little information about the performed machine learning—feature extraction and model creation and validation. The system was utilized by the creators in several studies, e.g., to assess employee satisfaction (happiness, activity, and stress) [81], a person's creativity [82], and even personal moral values [83]. However, in most of these studies, the authors took part in the study as subjects, which can lead to bias and can question the quality of the results [84].

Tripathi et al. proposed an EmoWare framework for personalized video recommendations [85]. They utilized the reinforcement learning approach to extract users' individual behavior and a deep-bidirectional RNN to predict relevant movies. The system has not been validated in real life. In general, most proposed systems do not leave the laboratory walls, i.e., they are not validated in daily life. There are also cases when the field validation is performed improperly. For instance, Fortune et al. [86,87] claimed that facial- and EEG-based emotion recognition of employees in real-time is possible. In fact, in three points of the day (before the work, in the middle of the day, and after the work), the authors showed short affective movies to the employees and measured their EEG and EDA at those moments. Later, reactions to the stimuli were classified into positive vs. negative states. Hence, the authors recognize emotions in response to the consumed stimuli. The employees' emotions were not connected with the work-related conditions or environment, or type of job they perform. The authors' claims of recognizing emotions at work were exaggerated, as the context of the work is barely relevant here.

Hernandez et al. looked over some of the recent emotion recognition commercial applications in market research, human resources, transportation, defense and security, education, and mental health areas [88]. They discussed several issues of existing systems, e.g., (1) systems describe their functionality insufficiently, often hiding the technical limitations; (2) a limited amount of data exists that can be used to train models; (3) users may over-rely on the systems, which can lead to the physical or emotional injury; (4) privacy, freedom, and other human rights may be abused by using/applying such systems; (5) and more generally, emotions do not have agreed definition and are difficult to label. The authors propose the ethical guidelines that should help mitigate the recognized issues.

Huge corporations offering services to millions of people noticed the importance of including affective aspects in their products. For example, Netflix and Spotify offer search engines that consider how the content affects the emotional state of the user [89,90]. The next step is probably to automate such solutions, e.g., by creating personalized playlists tailored to our affective state at a given moment [91].

## 6. Futures of Emotion Recognition

Emotion recognition studies are most likely to be continued in three main directions, i.e., ER from facial expressions, speech, and physiology, because each approach has its unique advantages and applications. The camera-based ER from the face is comfortable and accurate enough when sitting in front of a computer, although it can be deceived by controlling facial expressions. The ER from speech is an ideal solution for voice assistants and human–robot communication that is soon a reality. The personalized models could be stored locally on the devices, thus improving the privacy and security of such a solution. The ER from physiology has the most significant potential to become the most popular and broadly applied approach. Comfortable and user-friendly wearables monitoring our vital functions in almost any condition are available and already pervasive in our life. At the same time, physiology is the most difficult to manipulate; hence, it can be considered ground truth for other approaches. Another advantage of physiology is the possibility of noncontact measurement, e.g., a camera analyzing the change in the skin texture or color [92,93] or radio-frequency analysis for breath and heart rate detection [94].

Several challenges need to be tackled to enable precise and reliable affective reasoning in daily life. Currently, the most significant imperfection of sensors measuring physiology is the inaccuracy of measurement in motion. An unreliable measurement undermines any further effort to recognize emotions during movement. These imperfections can be partially reduced with advanced signal processing and transformation methods. However, such a fundamental issue should be fixed at the sensor level, i.e., new ways of measurements prone to motion have to be developed. Simultaneously, a universally accepted protocol of a new device validation against medical-level hardware should be proposed.

In addition, wearable devices should become even smaller, even more convenient, and even more useful for users while offering long battery life, convenient data transfer, and reliability. Particularly prominent directions seem to be smart clothes, smart prosthesis [95], and other everyday items that can be made *smart*. Until then, most probably, smartwatches will be the primary source of physiological data and the foundation of affective field studies. They are cheap, useful for users, have many sensors (BVP, ACC, and GYRO), allow for integration with systems through a custom-made application and, above all, are already omnipresent. Their only disadvantage is the lack of an EDA sensor, which can be easily fixed.

In the signal processing and transformation domain, more algorithms for artifacts and outliers removal and reconstruction of signals recorded in motion are necessary. Especially promising are the wavelet-based methods that extract affective patterns from the signals.

In the machine learning field, new deep learning architectures customized for affective studies can be developed (such as in the case of the StresNet). Such architectures would preferably reflect complex relations across emotions and their personal character. While very little has been investigated so far, a multilabel approach to emotion classification could potentially reflect the natural coexistence of emotions in real life. Further research on model generalization and personalization is also necessary. Particularly, new strategies for combining the general underlying human patterns with the significant individual differences are required.

Large and high-quality data sets must be collected to develop better predictive models. This, in turn, requires simple and accurate methods to annotate the signal with the affective label during everyday life. The current solutions include developing and validating short, yet rich in information, self-assessments that may be completed on the smartwatch or smartphone, such as ESM [96] or EMA [97]. Another option is to assist the participant with a pretrained model that recognizes states of arousal or intense emotions [27,98].

The ethical part of affective studies is a couple of years behind the technological advancements and still insufficiently discussed. We need more analysis, debate, and legal actions to ensure that affective technology is safe for the people and is not abused.

## 7. Conclusions

In recent years, we have experienced a number of breakthroughs in sensors and machine learning domains that make it possible to move the affective studies out of the lab into real life. Although there are still challenges to overcome, the first systems are already being used in everyday life. Further improvements, however, are necessary to enable the development of more precise and reliable systems.

This paper discusses the current state of the art in sensors, signal processing and transformation, and deep machine learning areas. Commercially available wearables and other devices used to capture traces of human emotions are presented. The role of signal processing and transformation in emotion recognition is explained with examples. The most prominent deep learning architectures and various techniques to improve model efficiency are extensively discussed. Finally, the research directions with the highest opportunity to improve emotion recognition are suggested.

**Funding:** This work was partially supported by the National Science Centre, Poland, project no. 2020/37/B/ST6/03806; by the statutory funds of the Department of Artificial Intelligence, Wroclaw University of Science and Technology; and by the Polish Ministry of Education and Science–the CLARIN-PL Project.

**Acknowledgments:** The author would like to thank members of the Emognition research group for their contribution to this article.

**Conflicts of Interest:** The author declares no conflict of interest.

## Acronyms

The list of acronyms used in this article:

**Signals, Sensors, and Data From Wearables**

| Acronym | Full Name |
| --- | --- |
| ACC | accelerometer |
| AL | ambient light |
| ALT | altimeter |
| AT | ambient temperature |
| BAR | barometer |
| BP | blood pressure |
| BVP | blood volume pulse |
| CAL | calories burned |
| ECG | electrocardiogra(ph/m) |
| EEG | electroencephalogra(ph/m) |
| EMG | electromyogra(ph/m) |
| GSR | galvanic skin response |
| GYRO | gyroscope |
| HR | heart rate |
| HRV | heart rate variability |
| IBI | interbit interval |
| MAG | magnetometer |
| MIC | microphone |
| PPG | photoplethysmograph |
| PPI | peak-to-peak interval |
| RRI | R-R interval |
| RSP | respiration rate |
| SKT | skin temperature |
| SpO2 | blood oxygen saturation |
| STP | no. of steps |

| | |
|---|---|
| TERM | termometer |
| UV | ultraviolet |

**Deep Learning Architectures and Signal Processing**

| Acronym | Full Name |
|---|---|
| Bi- | bidirectional- |
| CNN | convolutional neural networks |
| DAE | denoising autoencoder |
| DTD | dynamic threshold difference |
| FCN | fully convolutional networks |
| FFA | fast Fourier transform |
| GAN | generative adversarial networks |
| GRU | gated recurrent unit |
| ICA | independent component analysis |
| LSTM | long short-term memory |
| MCDCNN | multichannel deep convolutional neural networks |
| MLP | multilayer perceptron |
| ResNet | residual networks |
| RNN | recurrent neural networks |
| SPARE | spectral peak recovery |
| StresNet | spectrotemporal residual networks |
| VGG | visual geometry group |
| WT | wavelet transform |

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
