# Peer review of "Bringing Emotion Recognition Out of the Lab into Real Life: Recent Advances in Sensors and Machine Learning"

_electronics, doi:10.3390/electronics11030496_

Round 1

Reviewer 1 Report

This review paper shows a reasonable motivation for different life applications. Also, it shows the state-of-the-art schemes and sensors. However,  there are some minor issues:

Introduction

  1. Electronics created small and effective sensors that capture images/video. It is better to write videos. Line 28.

Sensors and Devices

  1. Are there any new references rather than [24, 25], especially, with reference 25 which is 2017? Line 47.
  2. It is recommended to write in Line 49. Primarily, ECG, BVP, and EDA signals are used [5,9,12].
  3. It is recommended to rephrase the following in Line 61. Other devices are better suited for studying emotions, others for meditation, yet others for studying sleep, and yet others for monitoring physical activity [30].

Facial Emotion Recognition

  1. Line 95. Ideally, the face would be turned towards 96 the camera, without any obstacles (hair, eyeglasses, face mask). There are many machine learning approaches that could overcome the occlusion problem (eyeglasses, face mask), and it could be used with this type of sensor.

End-to-End Deep Learning Approach

  1. The reference in line 313 is not mentioned. Schmidt et al. utilized CNN to build an end-to-end architecture [?].
  2. In line 321, it should be written Sun et al., not Sun. Sun used Residual CNN as end-to-end architecture in the speech emotion recognition task [65].

Representation Learning

  1. In line 535, it should be written as to have not has. They showed that stress detection with translated features has the competitive performance to stress detection with the original signals.

Existing Systems

  1. In line 417, it is recommended to be written as claimed not to claim. For instance, Fortune et al. [82,83] claim that facial- and EEG-based emotion recognition of employees in real-time is possible.

Reviewer 2 Report

Emotion recognition is an important research topic, which can be widely applied to daily activity. In this paper, the authors focus on reviewing recent advances in emotion recognition, and bring the technology out of the lab into real life. It is definitely a very interesting paper to read. The paper is well-outlined: first the authors discuss the hardware including sensors and devices, then move onto the soft part, such as signal processing, and machine learning models, and last is the current existing systems and future directions. Though different hardware/software topics are included in different sections, the majority of the paper is discussing deep learning models in emotion recognition, which is quite understandable as deep learning is the current trend in many areas of research. Here are my detailed comments or questions regarding this paper:

1. There are many acronyms in the paper not spelled out, particularly in section 2. Maybe the authors can consider making a list or an appendix for explaining the acronyms. Table 1 is an effort to show the devices and sensors, and it is great to know that the authors actually tested many of the sensors in the table. However the way Table 1 is organized is not clear enough for read and comparison. The content in "Sensors" column and "Other data" column are not sorted, which makes it very hard to know which sensors the devices have or have not. Another way to organize Table 1 is to list all available sensors and use check marks to indicate if one device has the sensor or not. 

2. In the section 2 "Sensors and Devices", the authors discussed Emotion Recognition in physiology, face and speech three aspects. While in section 4 "Machine Learning Models and Techniques", the discussion is based on the structure of the networks, such as ResNets, LSTM, CNN and FCN etc. For deep learning, supervised training requires large scale of labelled data. So I think it is necessary for the authors to introduce the format of the training set, its quantity, quality and labels derived from the sensor output. Table 2 listed network architecture for different emotion recognition tasks. Obviously there are similar/different ways of labelling between works, e.g. facial/speech emotion can be categorized into four classes or seven classes. The organization of the content in column "Classification/Regression problem" is not easy for readers to identify classification similarity/difference. The authors could re-organize the table to make it clearer to read.  

3. As the authors discuss deep learning emotion recognition method based on network structure in section 4, my question is: does one network structure apply to all different physiology, face or speech emotion recognition tasks? If yes, then emotion recognition is no special task but a generic use case of deep learning for classification. If not, then the authors should emphasis on how each network algorithm is customized for the emotion recognition task. 

4. Line 313 has a [?] for the citation.

5. The title of this paper is "Bringing emotion recognition out of the lab into real life", while in the paper, I don't see a lot of emphasis on the real-life application of all the reviewed works. Maybe the authors can mention for each work, if it is in lab or in the field. 

In general, this is a significant work of reviewing emotion recognition in physiology, face and speech from hardware devices to algorithms, and existing systems. There are many topics and methods to cover and the authors focus on the state-of-the-art deep learning methods. With further revision, it is good to publish. 

Reviewer 3 Report

The author reviews the recent progress in emotion recognition using the state of the art in machine learning and sensing technology.

The report is mainly qualatative and the authors should provide some examples of the quantative performance results of the described systems.

A discussion on the general direction of the ER research would be beneficial. What are the competing directions and threats to each?

A thorough review of the literature is provided. The Acronyms used should be defined e.g. PPG etc.

Authors could consider a figure to show the relatioship between the main technology approaches.

The machine learning technology behind the systems (e.g. LSTM) is described with concise but sufficient detail to be of interest and use to the reader.

There are some issues that should be verified:

Check the unknown reference After Ref. 63 on line 313

Round 2

Reviewer 3 Report

I am satisfied that the author has made the improvements as suggested. I recommend the acceptance of the mansucript.